

# Estimating the uncertainty of middle-atmospheric temperatures retrieved from airborne Rayleigh lidar measurements

Stefanie Knobloch[1], Bernd Kaifler[1], Markus Rapp[1,2]

[1]Deutsches Zentrum für Luft- und Raumfahrt, Institut für Physik der Atmosphäre, Oberpfaffenhofen, Germany
[2]Meteorological Institute Munich, Ludwig-Maximilians-Universität München, Munich, Germany

*Correspondence to*: Stefanie Knobloch (stefanie.knobloch@dlr.de)

**Abstract.** Possible uncertainties of lidar measurements of middle-atmospheric temperatures, measured with the novel airborne Rayleigh lidar system ALIMA, are investigated on the basis of data from the SouthTRAC-GW campaign in September 2019 and corresponding simulations of photon counts of the ALIMA system. We evaluate uncertainties due to the attenuation by
Rayleigh extinction and ozone absorption, (signal-induced) photon noise, the photon background, and the nonlinearity of photon counting detectors. Ozone absorption induces an altitude-dependent cold bias in the retrieved temperatures of 2 K between 25 km to 55 km. Rayleigh extinction introduces a similar uncertainty of 2 K below 25 km that can be decreased by a suitable correction. Photon noise can introduce uncertainties of ±25 K at high altitudes (above 70 km) for high temporal resolutions (1 min), but on average the photon noise influences the temperature by only 1 K to 2 K at 70 km and decreases
downwards. Uncertainties related to the photon background and the nonlinearity of the detectors, with a dead time correction applied, play a minor role in the temperature uncertainty. The analysis of the photon background in the ALIMA measurements of six research flights of the SouthTRAC-GW campaign proves the assumption of a constant photon background with altitude as well as the Poisson distribution of the photon counts. The airborne operation of ALIMA is advantageous as the high flight altitudes reduce the Rayleigh extinction by up to 17 % and thus result in higher signal levels compared to a ground-based
operation. Overall, our analysis reveals that temperatures can be retrieved from ALIMA measurements with a remaining uncertainty of ≤ 1 K if all known biases are corrected.

## 1 Introduction

Rayleigh lidar (light detection and ranging) systems have been used since nearly 40 years to observe the middle atmosphere. The detected light backscattered by air molecules is proportional to the atmospheric density and enables the retrieval of
temperature (Hauchecorne and Chanin, 1980). The temperature is calculated via hydrostatic integration based on the hydrostatic equation and ideal gas law. The lidar return signal decreases exponentially with altitude as does the atmospheric density. Additionally, the lidar signal decreases with the range squared since scattered light forms spherical waves. The scientific interest to study various dynamical processes of the middle atmosphere, e.g. gravity wave dynamics (e.g. Lindzen, 1982; Fritts and Alexander, 2003; Fritts et al., 2006; Heale et al., 2014), needs efficient lidars. Present research of e.g. small-





scale gravity waves, wave breaking or polar mesospheric clouds (e.g. Kaifler et al., 2013; Kaifler et al., 2018; Fritts et al., 2019; Kjellstrand et al., 2020) requires lidar measurements of high temporal and spatial resolution up to the resolution limits given by noise. The high technical demands that must be met to achieve a sufficient sensitivity makes mesospheric lidar observations challenging. For instance, powerful lidars, large telescopes and efficient detectors are required.

Ground-based lidars are able to provide high resolution observations. However, the stationary operation only allows for the 35    investigation of dynamical processes that propagate through the laser beam in time series of vertically resolved profiles (Dörnbrack et al., 2017). Examples of powerful lidars are the Arctic Lidar for Middle Atmosphere Research (ALOMAR) in Norway (von Zahn et al., 2000), the operational ground-based lidars of the observatories at the Haute Provence Observatory, France (e.g. Hauchecorne and Chanin, 1980; Hauchecorne et al., 1987, Keckhut et al., 1993) and at the Maïdo observatory, La Reunion (e.g. Baray et al., 2013; Keckhut et al., 2015), or the Compact Rayleigh Autonomous Lidar (CORAL) in Tierra del 40    Fuego, Argentina (Kaifler et al., 2020a; Kaifler and Kaifler, 2021; Reichert et al., submitted).

An airborne lidar facilitates the probing of different atmospheric conditions within a large geographical area and allows targeted observations of atmospheric phenomena, e.g. probing the spatial and temporal variability of atmospheric gravity waves excited by a mountain ridge. The use of an airplane as measurement platform has the advantage of decreasing the distance between the lidar and the region of interest in the middle atmosphere because of the flight altitude. Following, an 45    airborne lidar can be less powerful compared to a ground-based lidar since the attenuation of the laser beam and its backscattered signal in the troposphere is largely decreased. The Balloon Lidar Experiment (BOLIDE) already demonstrated the advantage of a shorter distance between the lidar and the middle atmosphere, leading to a significantly higher data quality as compared to ground-based instruments of the same size (Kaifler et al., 2020b). The first airborne operation of a Rayleigh lidar took place in 2014 during the DEEPWAVE campaign (Bossert et al., 2015; Fritts et al., 2018). The more powerful 50    Airborne LIdar for Middle Atmosphere research (ALIMA) instrument, which is scope of this study, was operated for the first time in September 2019 onboard the German High Altitude and LOnge range (HALO) research aircraft within the framework of the SouthTRAC-GW campaign (Rapp et al., 2021). The SouthTRAC-GW campaign was conducted in September 2019 in Rio Grande, Tierra del Fuego, Argentina with the aim of studying the middle atmospheric gravity wave activity in the vicinity of the Southern Andes, Drake Passage and Antarctic Peninsula. For a detailed overview of the SouthTRAC-GW campaign and 55    its objectives we refer to Rapp et al. (2021). First scientific results based on measurements with the ALIMA system can be found in e.g. Rapp et al. (2021) and Dörnbrack et al. (2020).

The middle atmosphere is optically thin in the visible part of the electromagnetic spectrum. Nevertheless, the small optical depths cause significant Rayleigh extinction and absorption by ozone which results in a non-negligible attenuation of the laser pulses propagating through the middle atmosphere. Leblanc et al. (1998) analysed the effect of ozone absorption with simulated 60    lidar measurements, which were generated based on a monthly-mean temperature profile. They showed that a correction for ozone absorption is necessary for achieving a high accuracy of the retrieved temperature profiles. The explicit effect on the retrieved temperatures was first shown by Sica et al. (2000). They calculated the maximum deviation for seasonally averaged temperature profiles to be approximately 1 K to 2 K at a laser wavelength of 532 nm.

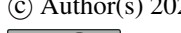



Potential uncertainties and biases of airborne Rayleigh lidar measurements and retrieved temperatures can originate from the
attenuation of the laser beam and its backscattered signal, the photon background, the photon noise, the nonlinearity of the
detectors or from inadequate assumptions made in the temperature retrieval, e.g. the improper selection of the seeding
temperature. In this work we focus on the determination of the accuracy and precision of the retrieved temperature profiles
and their effect on the analysis and interpretations of the data. We use simulated photon counts based on the characteristics of
ALIMA for the uncertainty analysis as the simulations allow us to investigate the impact of each source of uncertainty
separately.

The outline of this study is as follows: in the following Section 2 we describe how lidar measurements can be simulated and
how temperature is inferred from photon count profiles of Rayleigh lidar measurements. Section 3 presents the novel ALIMA
system and the used data sets. In Section 4 we then discuss ALIMA measurements and use corresponding simulations to
evaluated the relevance and magnitude of various sources of uncertainty. The results and insights of the paper are summarized
in Section 5.

## 2 Methods

### 2.1 Definitions

Within this paper we will use the following definitions: The *signal* describes the collected/incident and detected/counted
number of photons. Ideally, the photons comprising the signal are only laser photons backscattered by air molecules. However,
the signal further includes photons that originate from the photon background and the photon noise. The *photon background*
describes all collected and detected photons that originate from foreign light sources, e.g. the sun, moon and/or stars, and also
includes the noise caused by the dark current of the detectors. The photon background is assumed to be constant with altitude
(Keckhut et al., 1990). The *photon noise* and *signal induced noise* are a result of the signal itself. The photon noise is a statistical
fluctuation that is a consequence of the quantum nature of photons, which becomes noticeable in particular at low signal levels.
The signal induced noise is a source of non-linear noise that results from signal induced physical changes within the detector.
E.g. the heating of the semiconductor junction due to the current flowing across the junction in the avalanche photodiodes.
The *nonlinearity* of a detector describes the nonlinear response of a detector to incident light, i.e. the ratio of detected photons
versus incident photons depends on the rate of incident photons. A major contribution to the nonlinearity results from the *dead
time* of the detector which is a period immediately after the detection of a photon during which the detector is inactive and no
photons can be detected.

### 2.2 Simulating a Rayleigh lidar

We simulate raw photon count profiles, comparable to the counted photon profiles of a lidar, with the use of the Rayleigh lidar
equation (e.g. Fujii, 2005; Weitkamp, 2005):



$$P_R(\lambda, z) = \left(P_L(\lambda)\left(\sigma_{Ray,180}(\lambda)\frac{R_d}{k_B}\rho(z) + \sigma_{Mie}(\lambda)N_{aerosols}(z)\right)dz \ * \ e^{\left(-\tau_{Ray}(\lambda,z)-\tau_{o3}(\lambda,z)\right)} * \frac{A}{(z-z_L)^2} * \eta_{FOV}(z) + \right.$$


$$\left. P_{BG}\right) * \eta_{Trans} \tag{1}$$

where $P_R(\lambda, z)$ is the number of received photons based on the number of initially emitted photons by the laser $P_L(\lambda)$ with wavelength $\lambda$ at altitude $z$ and $z_L$ as the altitude of the lidar (in this case the flight altitude), $\rho(z)$ is the air density, $R_d$ is the gas constant of dry air, $k_B$ is the Boltzman constant and $N_{aerosols}(z)$ is the number density of aerosols at the altitude $z$. The laser beam is modified during its propagation through the atmosphere by scattering and absorption due to molecules and

aerosols. Therefore, $\sigma_{Ray,180}(\lambda)$ and $\sigma_{Mie}(\lambda)$ are the Rayleigh and Mie backscattering cross sections at the laser wavelength. The Rayleigh backscattering cross section is derived from the total scattering cross-section $\sigma_{Ray}(\lambda)$ as $\sigma_{Ray,180}(\lambda) = \frac{1}{4\pi}\frac{3}{2}\sigma_{Ray}(\lambda) = \frac{1}{4\pi}\frac{3}{2} * 5.16*10^{-31} \text{ m}^{-2} = 6.16*10^{-32} \text{ m}^{-2}$ (Bucholtz, 1995). The round-trip attenuation or extinction of the laser beam is given by the optical depth $\tau$. The major contributions to the attenuation at 532 nm wavelength are the Rayleigh extinction $\tau_{Ray}(\lambda, z)$ and the absorption by ozone $\tau_{o3}(\lambda, z)$:

$$\tau_{Ray} = 2\sigma_{Ray}(\lambda)\frac{R_d}{k_B}\int_{z_L}^z \rho(z')\delta z' \tag{2}$$

$$\tau_{o3} = 2\sigma_{o3}(\lambda)\int_{z_L}^z N_{o3}(z')\delta z', \tag{3}$$

with the total Rayleigh scattering cross section $\sigma_{Ray}(\lambda)$ and the ozone absorption cross section $\sigma_{o3}(\lambda)$ (Voigt et al., 2001) for the wavelength of the laser. In our simulations we assume that the extinction due to scattering at aerosols is negligible as the altitude range of interest lies between 30 km and 90 km which is assumed to be free of aerosols and well mixed. The factor

two in Eq. (2) and (3) originates from the roundtrip of the light through the atmosphere, since the light experiences the attenuation both during the upward and downward propagation. The number of received photons further depends on the solid angle $\frac{A}{(z-z_L)^2}$ of the receiving telescope of the lidar, with $A$ being the telescope collection area, the distance $(z - z_L)$ and the efficiency of the Field of View (FOV) $\eta_{FOV}$ which describes how well the lidar beam is aligned with the telescope, i.e. what fraction of the laser beam is visible in the telescope FOV. The received number of photons further depends on the efficiency

of the lidar $\eta_{Trans}$ as product of all efficiencies and transmissions of the optical components and detectors in the transmitter and receiver chain. Lastly, a photon background rate $P_{BG}$ is added to account for the sky background radiation and the dark current of the detectors.

Thus Eq. (1) can be expressed as a simplified, laser pulse-wise simulation (per emitted laser pulse) in units of photons:

$$P_R(\lambda, z) = \left(\frac{E\lambda}{hc} * \sigma_{Ray,180}(\lambda) * N(z) * dz \ * \ e^{\left(-\tau_{Ray}(\lambda,z)-\tau_{o3}(\lambda,z)\right)} * \frac{A}{(z-z_L)^2} * \eta_{FOV}(z) + \right.$$


$$\left. P_{BG}\right) * \eta_{Trans} * dt \ * f_{rep} \tag{4}$$

Equation (4) is evaluated for a certain number of laser pulses, given by the integration period $dt$ and the pulse repetition frequency $f_{rep}$. $E$ is the pulse energy and $\lambda$ the wavelength of the laser; $h$ is the Planck constant and $c$ the speed of light. $N(z)$




is the number density of air molecules, which is related to the air density by $\frac{R_d}{k_B}\rho(z) = N(z)$. The constants of this relation can also be expressed by the Avogadro constant $N_A$ and the molar mass of dry air $M_d$, since $\frac{N_A}{M_d} = \frac{R_d}{k_B}$.

Considering that a real lidar is not a noise-free system but introduces a Poisson-distributed noise/fluctuation to the counted photon signal due to the statistical behaviour of the quantum nature of photons (e.g. Gatt et al., 2007; Goodman, 2015), we also add a Poisson-distributed noise to the simulated photon count profiles. The simulated photon counts therefore include, besides the signal itself, the photon noise of the signal, the photon background and photons of the dark current of the detector. The photon counts are calculated as real numbers in Eq. (4), but are converted to integers before the calculation of the Poisson-

distributed noise to reflect the integral nature of photons. The extension of the photon count simulation to altitudes beyond 90 km is possible. But the change in atmospheric composition needs to be considered (Argall, 2007). Below 90 km, the atmosphere is well mixed and the atmospheric composition is therefore constant with altitude. Hence, the Rayleigh lidar photon counts can be directly related to the atmospheric density and number density:

$$\rho(z) \propto N(z) \propto P_R(\lambda,z) * (z - z_L)^2 \tag{5}$$

**2.3 Temperature retrieval**

By inserting the ideal gas law into the hydrostatic equation one can retrieve a temperature profile from a density profile. Therefore, one has to determine the density from the photon count profiles of a Rayleigh lidar measurement following Eq. (5). To ensure the proportionality between the photon count profiles and the atmospheric density the following steps are necessary: the preparation of non-ideal lidar signals includes a correction for the dead-time of the detectors, estimation and subtraction of

the photon background, range-correction, smoothing of the signal and re-binning of the count profiles to the desired vertical and temporal resolution. For the preparation, we follow the approach by Kaifler and Kaifler (2021). The temperature can afterwards be retrieved by assuming hydrostatic equilibrium (Hauchecorne and Chanin, 1980; Leblanc et al., 1998), i.e. the temperature can then be hydrostatically integrated from top to bottom as:

$$\int_{z_0}^{\infty} p(z)dz = -\int_{z_0}^{\infty} \rho(z)g(z)dz = -\left(\int_{z_0}^{z_{init}} \rho(z)g(z)dz + \int_{z_{init}}^{\infty} \rho(z)g(z)dz\right) \tag{6}$$

with the ideal gas $p = \rho R_d T$ to solve for the temperature. The integral of Eq. (6) ranges from infinity to zero. Since it is impossible to measure at infinity, the integral is split into two parts: an upper integral ranging from infinity to $z_{init}$ and a lower integral ranging from $z_{init}$ to $z_0$. It is then assumed that the upper integral equals the seeding condition $\rho_{init}T_{init}$ at $z_{init}$. Discretizing Eq. (6) leads to:

$$T(z_i) = \frac{1}{\rho(z_i)}\left[\rho_{init}T_{init} + \frac{M}{k_B}\left(\rho\left(z_i + \frac{\Delta z}{2}\right)g(z_i)dz + \sum_{z_{i-1}}^{z_{init}+1}\rho\left(z_i + \frac{\Delta z}{2}\right)g(z_i)dz\right)\right] \tag{7}$$

with $i = [1, ..., i, ..., bottom]$ increasing downward; the density $\rho\left(z_i + \frac{\Delta z}{2}\right) = \sqrt{\rho(z_{i-1})\rho(z_i)}$ as geometric average across a layer defined by altitudes $z_i$ and $z_{i-1}$; $M$ being the product $M_d * m_u$, with the unified atomic mass unit $m_u = 1.66 * 10^{-27}$ kg and the gravitational acceleration $g(z_i)$, computed from Eq. (17) of the U.S. Standard Atmosphere (1976). The here used density illustrates not the absolute air density but a relative air density that is proportional to the lidar photon counts (Eq. (5)).




The altitude $z_{init}$ at which the integration is started is determined as the highest altitude where a certain signal-to-noise ratio

(SNR) is first undershoot and the photon counts are larger than 10 counts per 100 m bin. The exponentially downward increasing SNR is calculated as

$$SNR = \frac{P_R - P_{BG}}{\sqrt{P_R}} = \frac{P_{signal}}{\sqrt{P_R}} \tag{8}$$

where the photon signal is obtained as difference between the total received photon counts $P_R$ and the photon background $P_{BG}$. The latter is evaluated in an altitude range, where the signal is insignificant, typically above 110 km. Subsequently, the

temperature integration is started at the determined altitude with an a-priori temperature value $T_{init}$, which has to be taken from another source, e.g. nearby satellite observations. These a-priori temperatures may only accord to a varying degree with the actual temperatures at the time of the lidar sounding because of the different temporal resolution, their geographic validity or because the measurement took place at a different point in time.

We execute the temperature retrieval with a time-pyramid approach (Kaifler and Kaifler, 2020): Firstly, the photon count

profiles are temporally binned to a nightly mean photon count profile and the respective temperature integration is initialised with the a-priori value $T_{init}$. The nightly mean photon count profile has the greatest SNR compared to higher temporal resolutions due to the smoothing of the noise. Hence, the initialisation, based on a certain SNR value, happens at the greatest possible altitude. Hereinafter, the photon count profiles are temporally binned in an iterative approach to ever increasing temporal resolutions. At each time, the temperature integration is initialised with temperatures derived from the previous lower

temporal resolution. This approach provides seed temperatures that are closer to the actual temperature values for the integration of high temporal resolution photon count profiles.

**Table 1.** ALIMA instrument characteristics as used in the simulation

| Laser pulse energy $E$ | 140 mJ |
|---|---|
| Laser wavelength $\lambda$ | 532 nm |
| Laser pulse repetition frequency $f_{rep}$ | 100 Hz |
| Altitude bin resolution $dz$ | 100 m |
| Integration period $dt$ | 10 s |
| Telescope area $A$ | 0.196 m$^2$ |
| FOV efficiency $\eta_{FOV}$ | 1 |
| Transmission efficiency $\eta_{Trans}$ | 0.239 |
| ▪ Surfaces | 0.477 |
| ▪ Quantum efficiency of detector | 0.5 |





## 3 Instruments and data sets

### 3.1 ALIMA

The ALIMA system is a novel middle atmosphere Rayleigh lidar developed by the Deutsches Zentrum für Luft- und Raumfahrt (DLR) for airborne operation on the German research aircraft HALO. The pulsed 532 nm laser beam, produced by a frequency-doubled diode-pumped Nd:YAG laser, exits the aircraft through a window in the ceiling (Fig. 1). Photon count profiles in the altitude range from approximately 20 km to 90 km are obtained by collection and detection of backscattered photons from the laser beam. The backscattered laser light passes a mechanical chopper which blocks the fraction from the lowest altitudes in order to protect the detectors from saturation by very strong signals. The beam is then split into three elastic channels: two channels (far channel 'ch0' and mid channel 'ch1') equipped with avalanche photodiodes (APD) and one channel (low channel 'ch2') with a photomultiplier tube (PMT) as detector. The splitting of the returned photon signal into different channels with the partitioning of 90%:9%:1% is necessary because the strength of the lidar signal changes by more than eight orders of magnitudes from 20 km to 90 km and thus exceeds the dynamic range of a single detector. The number of backscattered photons arriving at the aircraft's window does not equal the number of photons that are detected. Losses occur due to imperfect optical coatings of the window, telescope mirrors and receiver optics. Taking into account the quantum efficiency of the detectors (e.g. quantum yield of the photocathode of a PMT) being smaller than one, the detected light intensity is reduced compared to the incoming light intensity according to the efficiency value $\eta_{Trans}$. In total, within the ALIMA system, the

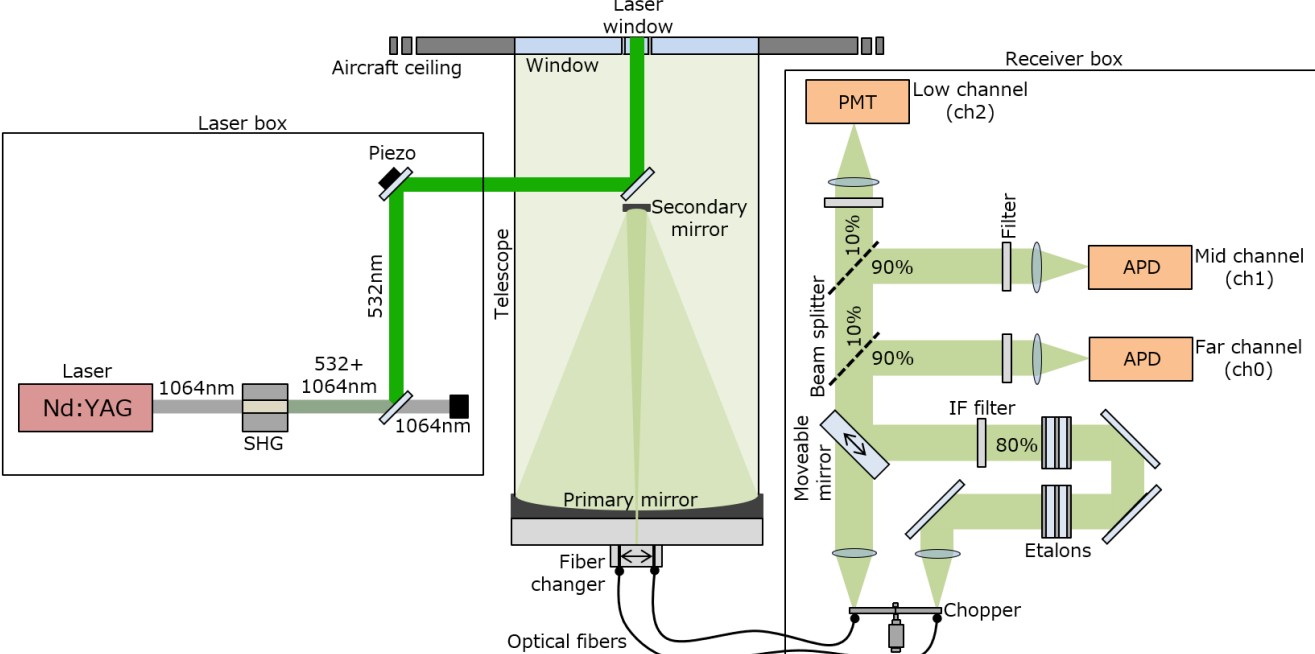

**Figure 1:** Simplified schematics of the ALIMA system as used during the SouthTRAC-GW campaign onboard the HALO aircraft. The set-up allows for switching branches in the receiver box for measurements in daylight (with etalons) and darkness (without etalons).





intensity of the backscattered light is reduced by 76.13 % due to the various losses (Table 1). A potential nonlinear behaviour of the detectors may cause an additional reduction of the detected photon counts (reduced photon detection efficiency). Nonlinearity causes the number of incident and counted photons to be not proportional. One contributing factor is the potential of pulse-pileup as consequence of the deadtime of the detectors. Another factor is the signal-induced noise. Typically, the

nonlinear behaviour cannot be neglected anymore if the photon count rates exceed approximately 1 MHz. Experiments in our laboratory showed that the commonly used procedures for correcting dead-time effects work well for the detectors used in ALIMA (dead time < 22 ns) until about 5 MHz. For that reason, maximum count rates are generally limited to 5 MHz with the help of the chopper and through gating of the detectors. In the postprocessing, the received photon counts are binned into discrete predefined time intervals with an integration period of 10 s and over a predefined altitude range of 100 m. In order to

improve the SNR, the received photon counts are vertically smoothed over 1500 m.

One big advantage of the airborne operation at upper tropospheric or lower stratospheric flight altitudes compared to a ground-based operation is that the lidar measurement is not influenced by clouds in the troposphere. Furthermore, the lidar is located closer to the probed volume, the middle atmosphere. The lidar return signal decreases (i) proportionally to the exponentially decreasing air density with altitude and (ii) with the altitude squared because the photons of the laser beam are scattered as

spherical waves. In addition, we obtain a stronger backscattered photon signal than with a ground-based system of similar power and efficiency because of the diminished attenuation of the laser beam by tropospheric Rayleigh extinction (Kaifler et al., 2020b).

Even though we expect to obtain lidar return signals only up to around 100 km, the ALIMA data acquisition records photons counts for altitudes up to 300 km. The included thermospheric altitude range is used to precisely measure the photon

background. The photon background is evaluated between 125 km to 190 km for the far channel and between 115 km to 180 km for the mid and low channels.

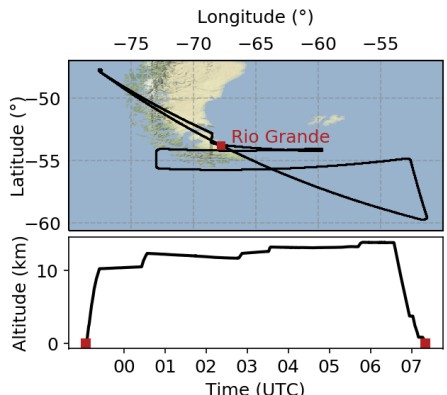

**Figure 2:** Flight track and altitude of the research flight ST08 of the SouthTRAC-GW campaign on 11/12 September 2019 over the southern tip of South America and the Drake Passage.



## 3.2 Data used in the lidar simulation

In this study, we simulate the raw photon counts $P_R(\lambda, z)$ based on the instrument characteristics of ALIMA listed in Table 1 for the flight ST08 of the SouthTRAC-GW measurement campaign. The research flight ST08 (Fig. 2), was conducted from 11

September 2019, 23:05:36 UTC to 12 September 2019, 07:21:14 UTC with the objectives to study orographic gravity waves, deep propagation, wave breaking, secondary gravity waves and the refraction into the polar night jet and along the gravity wave belt at 60°S. The atmospheric input of $N(z)$ for the simulation of photon counts using Eq. (4) is based on ERA5 temperature data. The four-dimensional ERA5 dataset is interpolated in space and time from hourly resolved fields (0.25° x 0.25° x 137 levels) to the coordinates of the ST08 flight track with a resolution of 10 s in time (can be translated to

approximately 2 km in horizontal space depending on the speed of the aircraft) and 100 m in the vertical. The smallest scales of gravity waves that can be represented in the lidar simulation are therefore limited by the resolution of the ERA5 data and limited to hydrostatic gravity waves due to the underlying hydrostatic model and the used data assimilation in the ERA5 model (Hersbach et al., 2020). Hence, we do not expect a perfect agreement between the measurements and model results. Difficulties in the ERA5 representation of upper stratospheric temperatures (Simmons et al., 2020) may also preclude an agreement

between the ALIMA measurements and the corresponding simulation.

Furthermore, we use two data sets for the simulation of ozone absorption: an ozone climatology and satellite observations. The ozone climatology by Fortuin and Kelder (1998) provides monthly zonal mean ozone values for 17 10° wide zonal bands at 19 pressure levels ranging from 0 km to 59 km, which are based on measurements from 30 ozonesonde stations around the world and solar backscattered ultraviolet (SBUV-SBUV/2) satellite observations. We use the climatologic ozone profile for September for the

meridional band 55°S to 65°S. Additional ozone satellite measurements for the time period of ST08 were obtained from the Earth Observing System (EOS) Aura Microwave Limb Souder (MLS), which measures radiance near 240 GHz for deriving ozone between 261 hPa to 0.001 hPa.

## 3.3 The lidar simulation

Figure 3 shows an example of photon count profiles measured by ALIMA and the corresponding simulation. The theoretical

transmission and quantum efficiency of 0.239 of ALIMA (Table 1) has to be reduced to $\eta_{Trans} = 0.08$ in the simulation of ST08 to obtain comparable photon counts. The reduced efficiency is likely a result of icing which occurred on the aircraft laser window during the flight. The limited vertical extent of the ERA5 data with a top of 78 km causes an abrupt transition between the simulated photon signal and the altitude bins above containing only the simulated photon background as compared to the smooth transition between signal and background in the ALIMA measurements. In order to avoid this discontinuity in the

retrieval, the initialization of the hydrostatic integration is limited to altitudes below 78 km. The temperature retrieval of the simulated photon count profiles is seeded with $T_{init}$ based on a mean temperature profile of the ERA5 temperatures along the flight track of ST08.





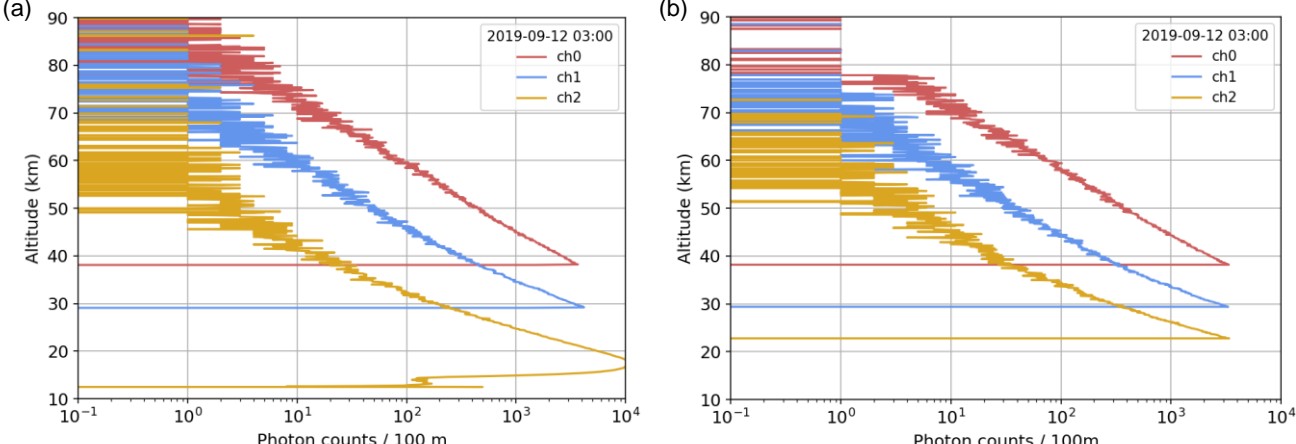

**Figure 3:** Channel-wise photon count profiles with an integration period of 10 s of (a) the ALIMA measurement of the flight ST08 at 03:00 UTC and (b) the respective lidar simulation based on Eq. (4). The horizontal stripes in the photon count profiles above approximately 50 km are the result of the logarithmic x-axis and the resulting insufficient representation of values alternating between zero and one (or two).

The temperatures and temperature perturbations of the flight ST08 retrieved from the measured ALIMA photon count profiles and the simulated photon count profiles as well as the ERA5 data are shown in Fig. 4. Despite the limitations of the ERA5 data (vertical extent, lacking gravity wave scales, etc.), an astonishingly similar temperature field is retrieved from the simulated lidar data with the described method. The inclusion of the photon noise in the simulation provoked an increase in amplitude of the temperature perturbations. Therefore, the perturbations appear much more similar to the ALIMA measurements in the retrieved temperatures from the lidar simulation than in the ERA5 data. The similarity between the simulation and ALIMA motivates us to study in the following different uncertainties based on the simulation and transfer the conclusions to ALIMA measurements.

## 4 Uncertainties in airborne Rayleigh lidar measurements

The simulation of the physics that affect lidar measurement allows the study of different sources of uncertainty within a controlled environment. In the following, uncertainties due to the temperature retrieval and its initialisation, the attenuation of the laser pulses, the photon background and nonlinearity of the lidar detectors, will be analysed and quantified based on the ALIMA measurements and the corresponding simulation results.

### 4.1 Temperature retrieval and initialisation

Ideally, the retrieved temperatures should equal the actual temperatures. However, differences between both temperature sets are present. Figure 5 presents the differences between temperatures retrieved from the simulated photon counts profiles version v1 of ST08 and the actual temperatures from the ERA5 data. Small integration periods, e.g. 1min, feature an altitude-dependent





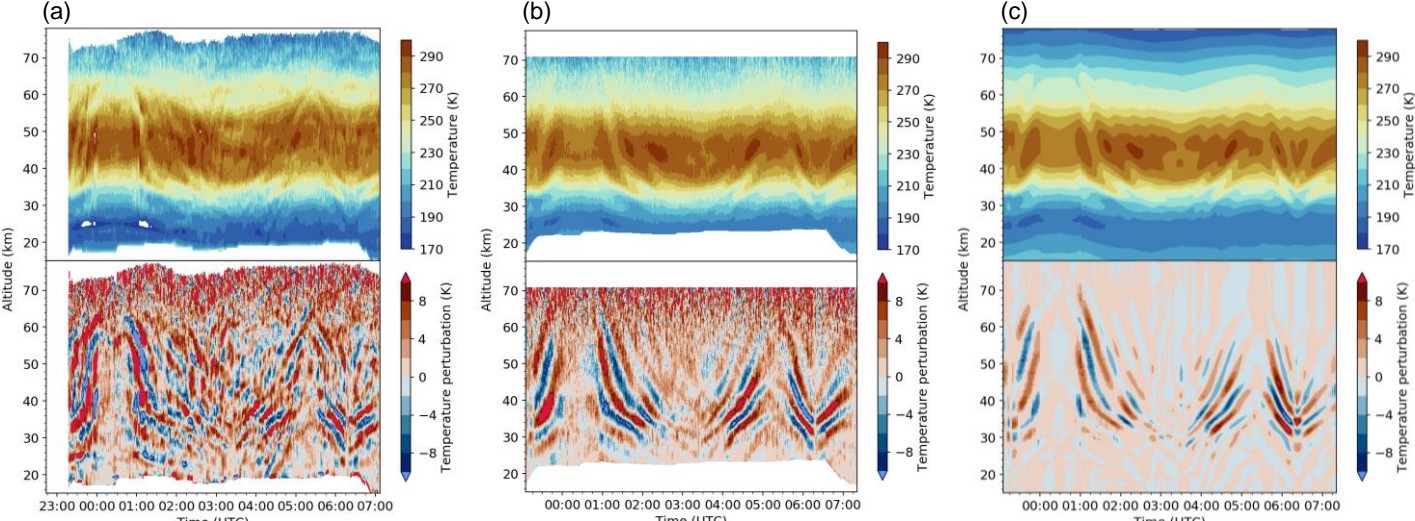

**Figure 4:** Timeseries of temperature and temperature perturbations (derived by subtracting a 30 min running mean from the temperature field) along the ST08 flight track of (a) retrieved from the ALIMA measurement, (b) retrieved from the lidar simulation based on ERA5 data and (c) ERA5 data. Temperatures are retrieved with an integration period of 1 min and vertical smoothening of 1500 m. Peak amplitudes of temperature perturbations smaller than -10 K are shown in bright blue and larger than +10 K are shown in bright red.

temperature bias up to 1 K in absolute numbers that is most pronounced in the stratosphere and upper mesosphere. The spread in temperature difference, approaching ±25 K at 70 km, is caused by the photon noise included in the photon counts, leading on average to a cold bias in the upper mesosphere (i.e. McGee et al., 1995; Thayer et al. 1997). The retrieved temperatures tend to be lower on average at altitudes with small signal levels for short integration periods because of the lesser smoothing of the photon noise. Since the photon noise follows a Poisson distribution, it has a greater probability to be larger than the

expected value for small signals due to the positively skewed tail of the distribution. Larger photon counts imply a greater density which leads to lower temperatures in the hydrostatic integration. The large spread of temperature differences in the stratosphere can be related to the to the increased uncertainty as a result of switching channels (approximately at 30 km and 40 km) and the accompanied sudden lesser photon counts. However, the on average cold bias of ≤ 1 K in the stratosphere is related to the performance of the hydrostatic integration since other uncertainties are either excluded or do not act in this

altitude range. Possible causes affecting the performance of the hydrostatic integration are the numerical error of the integration and errors related to the interpolation and smoothing of temperature and density profiles. Below 25 km, the temperature difference is less meaningful. Only a reduced number of photon count profiles and, therefore, number of temperature profiles, are included in the statistics because we truncate profiles where the rate of detected photons exceeds 5 MHz.

Due to the airborne operation of the lidar, larger integration periods, e.g. 4 min, become less useful due to the decreasing in

horizontal resolution (Fig. 5b). The temporal binning and accompanied averaging of photon count profiles does not take horizontal movement and flight manoeuvres into account. During climbs and dives of the aircraft, the observational volume is changed. Photon count profiles are thus affected by a different magnitude of uncertainty. During curves the aircraft rotates around the roll (longitudinal) and pitch (lateral) axes. The oblique attitude of the aircraft causes the laser beam to deviate from

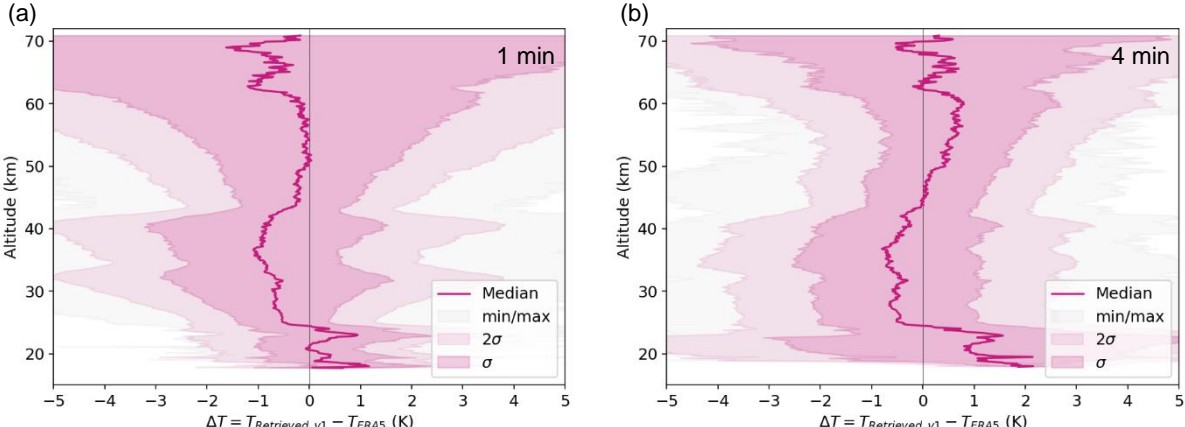

**Figure 5:** Difference between the retrieved temperatures based on the simulation (version v1 (Table 2)) of ST08 and the simulation input ERA5 temperature for (a) 1 min integration period and (b) 4 min integration period.

the vertical and cuts again through a different observational volume. The latter effect is not included in the simulation but
potentially increases the altitude dependent uncertainty in temperature as the path through the corresponding altitude ranges increases.

So far, we have not looked at the influence of choosing different a-priori values $T_{init}$ because the seeding temperatures were taken from the ERA5 data and thus equal the actual values used in the simulation. However, in more realistic cases and actual measurements, the true atmospheric temperature at the seeding altitude is not known and biases in the used a-priori values are
unavoidable. Figure 6 demonstrates the influence of different a-priori values $T_{init}$. The temperature integration of the simulated photon count profile is seeded with values with up to ±20 K difference from the actual temperature at 78 km, which includes realistic seeding errors. The bias is already bisected at 70 km and only one-tenth of the initial bias remains at 60 km (i.e. after 2.5 scale heights). The downward integration ensures that the contribution of the seeding error decreases quickly as the density increases. Below 50 km, the remaining error in temperature, due to an error in the $T_{init}$ value, is smaller than 1 K.

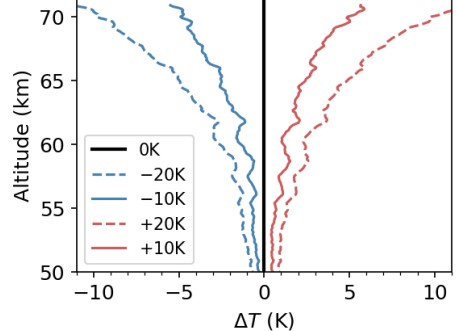

**Figure 6:** Mean temperature error between retrieved temperature with 10 min integration period from simulated photon count profiles (v1) without any seeding error (0K), ±10 K and ±20 K seeding error.



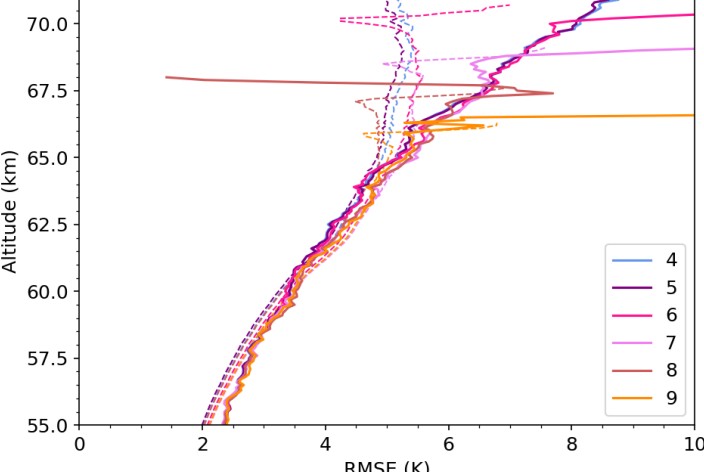

**Figure 7:** RMSE of the difference between retrieved (based on the simulation of ST08 and 1 min integration period) and actual ERA5 temperatures for six different SNRs of the far channel (ch0), while a SNR of 8 was used for the mid and low channels (ch1 and ch2). Dashed lines show the result for a simulation without the addition of Poisson-distributed photon noise. Large outliers at the top of the profiles arise due to varying number of profiles that go into statistics and should not be considered.

The altitude range where the integration starts, is characterized by low signal levels. The number of photon counts is typically between 10 to 30 counts per 100 m bin in this altitude range. Therefore, already a change by ±1 photon count related to photon noise has a significant influence on the resulting temperature. By the usage of the SNR for the determination of the seeding altitude, one wants to ensure that the signal is large enough to avoid a disproportionally large influence of the noise on the retrieved temperatures. Thus, the question arises "what is the ideal SNR to use?". Figure 7 shows that the root mean square

error (RMSE) differs only marginally, < 0.2 K, for different SNR thresholds. The only noticeable effect is that the higher the used SNR is for the far channel (ch0), the lower the temperature integration starts and thus the shorter the retrieved temperature profile. The spread of temperature differences decreases anyway downwards as the signal level increases. Therefore, using a larger SNR as threshold has no advantage if the uncertainty in retrieved temperature, elicited by the photon noise, should be reduced. SNRs between 1 to 3 are first reached at altitudes where about 10 photons per bin are counted (not shown).

Additionally, the SNR should not be too small because otherwise the temperature error due to the noise because too large. Hence, we use a SNR of 4 for the far channel for ST08. Retrieving temperature from simulated photon counts without the addition of the Poisson-distributed photon noise (dashed lines in Fig. 7) leads to a reduction of the RMSE of up to 3 K above 65 km compared to the case with photon noise. Below 65 km, the RMSE is rather similar between both cases. Hence, we can conclude that the uncertainty directly related to the photon noise can only be clearly delimited above 65 km. Below, the

uncertainty of the still small amount of photon counts becomes more import than the uncertainty of the photon noise.



## 4.2 Attenuation by Rayleigh extinction and ozone absorption

Two major effects attenuate the laser beam and its backscattered fraction as the light propagates vertically through the atmosphere: (i) the elastic (Rayleigh) scattering by air molecules, which influences the transmission of the electromagnetic radiation and (ii) the photodissociation of ozone in the stratosphere through the absorption of the electromagnetic radiation in
the wavelength regime of the Chappius continuum. Both related errors are small but still noticeable and will hence be quantified in detail.

The magnitude of attenuation through Rayleigh extinction depends on the flight altitude (Fig. 8a). Changes in flight altitude during airborne lidar measurements modify the sampling volume above the aircraft: the higher the aircraft flies the lesser air molecules are located inside the sampling volume and are able to interact with the laser beam. Therefore, the attenuation by
Rayleigh extinction has less influence the higher the flight altitude. For flight altitudes between 10 km to 14 km (Fig. 8a, black and blue lines), the total optical depth of approximately 0.025 to 0.035 can be translated to an absolute attenuation of ~2.5 % to ~3.5 %. A flight altitude of 0 km (ground-based lidar; Fig. 8a, grey line) results in a total optical depth due to Rayleigh extinction of approximately 0.21 to 0.22, which corresponds to an absolute photon count change of ~19 % to ~20 %. The magnitude of Rayleigh extinction becomes approximately constant above 25 to 30 km due to the exponential decrease of air
molecules with altitude. Since only the vertical change of the attenuation is decisive for retrieved temperatures, Rayleigh extinction significantly influences the lidar measurements below 30 km and it's effect becomes insignificant above 40 km.

**Table 2:** Major characteristics of the performed photon count simulations.

| Version | Simulation setting | Retrieval setting | Remark |
|---|---|---|---|
| v1 | No extinction included in photon count simulation | Rayleigh extinction correction off; no ozone absorption correction implemented | Assumption: similar to the case that Rayleigh extinction and ozone absorption are present and perfect corrections are applied in the retrieval |
| v2 | Only Rayleigh extinction included | Rayleigh extinction correction off or on; no ozone absorption correction implemented | |
| v3 | Only ozone absorption included | Rayleigh extinction correction off; no ozone absorption correction implemented | |
| v4 | Rayleigh extinction and ozone absorption included | Rayleigh extinction correction off or on; no ozone absorption correction implemented | Realistic simulation |





The influence of absorption by ozone on the laser beam and its backscattered fraction is independent of the flight altitude (Fig. 8b) because most ozone is located in the stratosphere well above the maximum flight altitude. With the usage of the ozone climatology, the maximum optical depth due to absorption by ozone is approximately 0.031, corresponding to ~3 % absolute

change in photon counts. This value is reached at 59 km in our simulation since the ozone climatology only reaches up to this altitude (Fortuin and Kelder, 1998). When incorporating MLS Aura satellite ozone measurements for the date of ST08 instead of the climatology, the attenuation can be determined to greater altitudes. The attenuation based on MLS Aura includes the diurnal and location dependent variability of ozone compared to the ozone climatology. The optical depth based on satellite observation reaches values of 0.025 to 0.031 at 80 km, which corresponds to ~2.5 % to ~3 % absolute change in photon counts.

The maximum difference in optical depths between the ozone climatology and satellite measurements is $\leq 0.01$ ($\leq 0.99$ % attenuation) and located above the stratopause at approximately 50 km. Above 40 km, the vertical change in the ozone optical depth is weaker when using satellite observations. Above 85 km, the optical depth based on satellite measurements increases due to the secondary ozone maximum in the upper mesosphere, lower thermosphere region. The strongest vertical change in the attenuation $\frac{d\tau_{O3}}{dz}$ is found around 35 km, therefore, we expect the largest temperature uncertainties in the lidar measurements

caused by ozone in this altitude range. Above, $\frac{d\tau_{O3}}{dz}$ decreases again. The relative density (Eq. (7)) is apparently larger in altitude ranges with increasing $\frac{d\tau_{O3}}{dz}$, resulting in lower temperatures at these altitudes (will be shown later). Despite the apparent

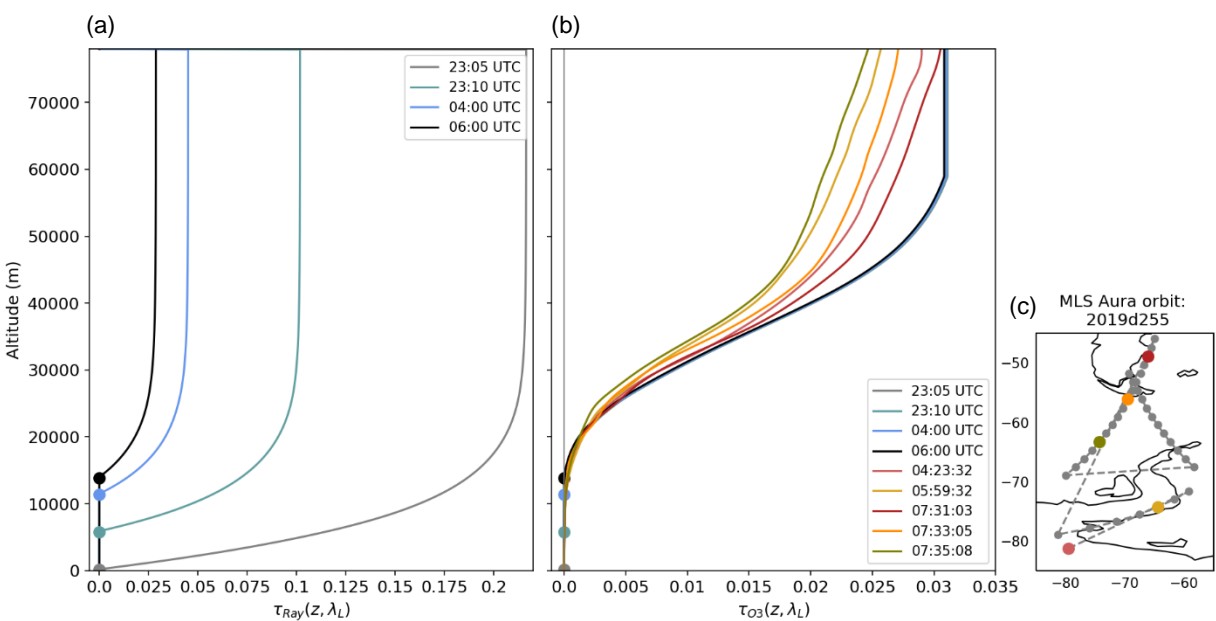

**Figure 8:** Optical depth $\tau$ based on (a) Rayleigh scattering for different time steps of the simulated ST08 flight (bluish colour) and (b) absorption by ozone based on the climatology for different time steps of the simulated ST08 flight (bluish colour) and MLS Aura observations for ST08 (reddish colour). (c) shows the track of the satellite and available measurement points. Calculations are based on Eq. (2) and (3), respectively.





differences when using the ozone climatology or satellite observations, similar absolute changes in photon counts are obtained (not shown). Hence, we consider it is sufficient to use an ozone climatology for a quantitative correction of the absorption by ozone in the temperature retrieval.

In order to determine quantitatively the impact of the attenuation by Rayleigh extinction and absorption by ozone on the retrieved temperatures, we performed several lidar simulations with different setups (Table 2): (v1) without any attenuation in the photon count calculation and no correction in the temperature retrieval, (v2) with attenuation by Rayleigh extinction in the photon count calculation, (v3) with attenuation by absorption by ozone in the photon count calculation and (v4) with both effects included in the simulation but still without any correction in the temperature retrieval. The absorption by ozone in the

simulation is based on the ozone climatology by Fortuin and Kelder (1998). The first simulation serves as reference case with the assumption that having no attenuation acting on the laser beam and its backscattered fraction and therefore no correction in the temperature retrieval produces the same result as having the attenuation acting on the laser beam and its backscattered fraction and a perfect correction applied in the temperature retrieval. The fourth simulation serves as the more realistic case in which the attenuation influences the airborne lidar measurement but no correction is applied in the temperature retrieval, which

therefore leads to deviations in the retrieved temperatures.

Figure 9 shows the temperature difference between the retrieved temperatures from the simulations v2, v3 and v4 and the reference case v1. Rayleigh extinction causes the retrieved temperatures to deviate on average up to -2 K below 30 km (Fig. 9a). We can account for this uncertainty by implementing a Rayleigh extinction correction in the temperature retrieval based on Eq. (2):

$$\tau_{Ray,corr} = \int_{z_L}^{z} 2\sigma_{Ray}(\lambda) \frac{P_{R(z\prime)}}{M_d} N_A \delta z' \qquad (9)$$

$$P_{R,corr} = \frac{P_R}{e^{-\tau_{Ray,corr}} / e^{-\tau_{Ray,corr}(z_{init})}} \qquad (10)$$

The correction uses the received photon counts for the estimation of the number density and the resulting attenuation by Rayleigh extinction. The correction reduces the temperature bias on average by 1 K at 22 km and 2 K at 18 km (not shown). Ozone absorption causes a temperature bias of up to -2 K between 20 km to 50 km (Fig. 9b). The peak temperature differences

at around 35 km coincide with the maximum $\frac{d\tau_{O3}}{dz}$, i.e. the largest uncertainty in temperature due to attenuation is caused by the largest vertical change in optical depth. The incorporation of attenuation by Rayleigh extinction and absorption by ozone without a corresponding correction causes the retrieved temperatures to exhibit an altitude-dependent cold bias (Fig. 9c).

The resulting lower temperatures are not intuitively understood since one might assume that the reduced photon counts would lead to a smaller air density and thus to higher retrieved temperatures. Including the attenuation in the photon count calculation

in Eq. (4) reduces the photon counts at all altitudes in and above ozone layers even though the attenuation is approximately


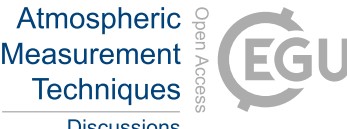

constant above a certain altitude. Hence, the photon count at $z_{init}$ influenced by any attenuation is always smaller than without any attenuation. Since the photon count at $z_{init}$ is determined as being proportional to the density and thus being consistent with the seeding temperature $T_{init}$, the actually diminished photon counts appear to be artificially increased. Therefore, lower

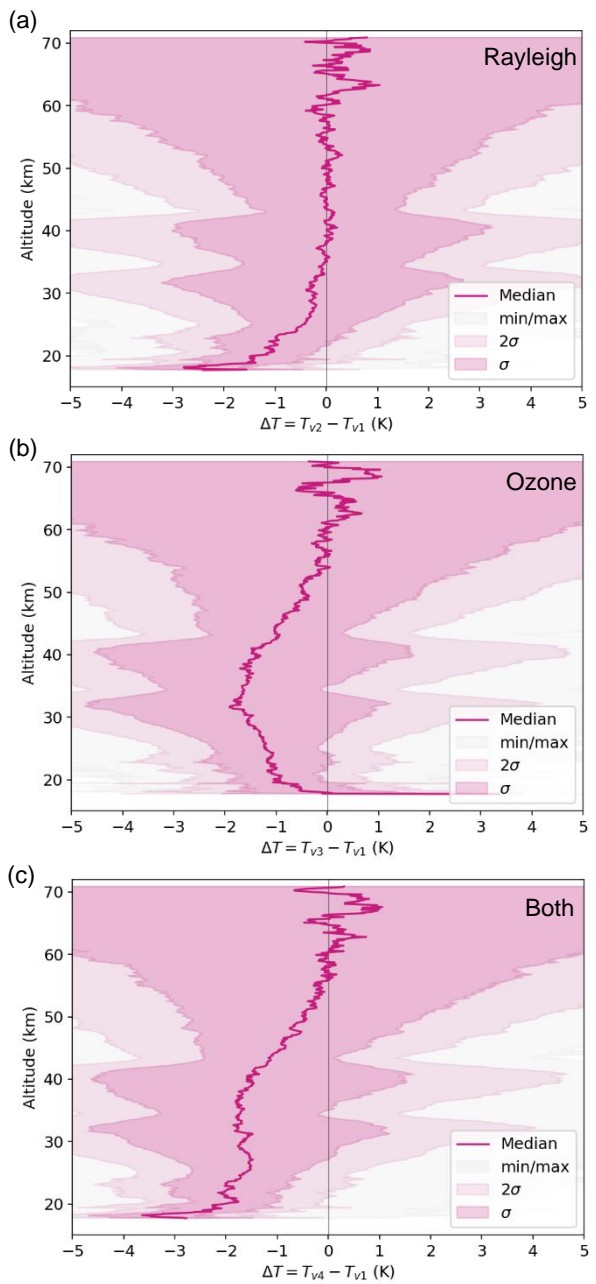

**Figure 9:** Similar as Figure 5a but for 1 min integration period and
(a) $T_{v2} - T_{v1}$ (b) $T_{v3} - T_{v1}$ and (c) $T_{v4} - T_{v1}$.





temperatures are retrieved from profiles of photon counts diminished by attenuation due to Rayleigh extinction and absorption by ozone.

A similar analysis of the influence of absorption by ozone by Sica et al. (2000) revealed temperature differences for middle and high latitudes of 1 K to 1.5 K at 25 km decreasing to 0 K at 47 km. Their study is based on seasonally averaged ozone number densities and temperature profiles. The ozone number density profiles used by Sica et al. (2000) decrease nearly monotonically with altitude, whereas the ozone number densities used in this study (ozone climatology and MLS Aura observation) contain an ozone maximum in the upper stratosphere. Hence, in our analysis the bias in uncorrected retrieved temperatures peaks at altitudes of the ozone maximum. Above, the bias decreases with altitude as in the analysis by Sica et al. (2000).

The determined magnitudes of temperature deviations due to attenuation by Rayleigh extinction and absorption by ozone reach values of up to 2 K. While the Rayleigh extinction influences only the lowest 30 km of lidar observations, absorption by ozone affects the lidar measurement over the entire stratosphere and lower mesosphere. This emphasises the necessity of a corresponding corrections for middle atmospheric lidar temperature measurements.

### 4.3 Photon background

The photon background $P_{BG}$ is assumed to be constant with altitude and can be calculated as the average photon count rate above altitudes where the returned lidar signal is negligible, typically above 110 km (Keckhut et al., 1990). The photon background included in the ALIMA photon count measurements and artificially added to the simulation originates (i) from foreign light sources, i.e. sun, moon or stars and (ii) from the dark current of the photon-counting detectors. The dark current results from the emission of electrons within the detector without incident light. In the case of the detectors used in ALIMA the dark current produces additional counts at a rate of 10 Hz to 30 Hz, depending on the particular detector. Based on lidar measurements, we cannot differentiate between the atmospheric photon background and the detector dark counts. Additionally, we cannot separate the photon background from the photon noise. The number of counted photons per time interval and, therefore, also the counts due to photon noise and photon background, follows a Poisson distribution bounded by zero.

Figure 10 shows the temporal evolution of the photon background during ST08, evaluated between 125 km to 190 km for the far channel and between 115 km to 180 km for the mid and low channels. The photon background peaks prior to the take-off (before 23:05 UTC), as well as the received photons in general, due to the residual twilight. Besides the noisy behaviour of the photon background due to the photon noise, a temporal modulation of the photon background of the far channel which coincides with changes in flight altitude and heading, is present in the ALIMA measurements. The geographic location and orientation of the airborne lidar relative to the source of the photon background influences this modulation.

The flight ST08 occurred in nearly full moon conditions and resulted in the second largest photon background of all research flights (Table 3). Flight ST09, which was conducted one day later, also approximately in full moon conditions, has the largest photon background of all flights. Furthermore, ST09 does not indicate a coincidence between the temporal evolution of the





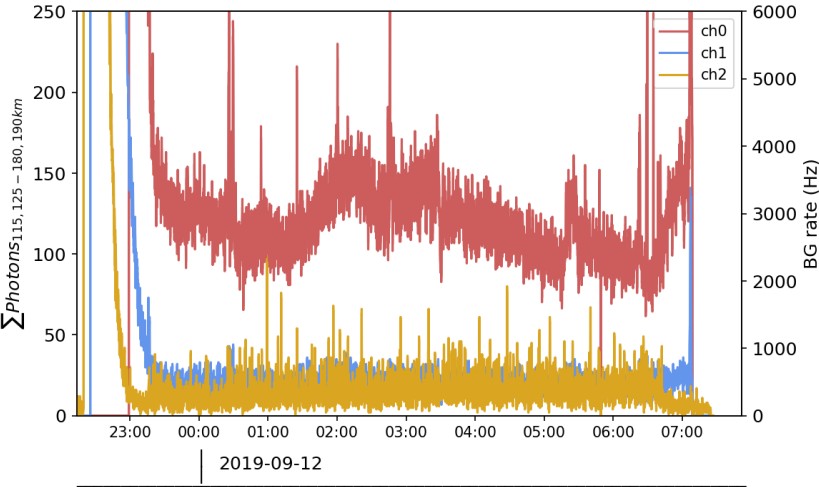

**Figure 10:** ALIMA measurement during ST08 of per channel sum of photons detected in the background altitude range and the resulting background rate. Photon counts are binned for a 10s temporal resolution.

photon background and changes in flight altitude (not shown). The issue of co-occurring icing of the laser window during ST09 is the most likely cause for the excess background.

Since the seeding altitude is not only determined by the SNR but also by the condition that the photon counts per bin are ≥10,
405 the photon background varies the photon counts by less than 1 % to 2 % at the seeding altitude for the night-time flights. The influence of the photon background decreases exponentially downwards with increasing signal. Therefore, the temperature uncertainty due to the photon background is negligible for the night-time operation of ALIMA.

**Table 3:** Statistics of the photon background from ST08, its simulation and 5 additional research flights from the SouthTRAC campaign
410 (ST09, ST10, ST11, ST12, ST14). Time periods of the detector switching on, take-off and landing are excluded from the calculation of the temporal mean. Statisitics are valid for 10 s integration period (1000 laser shots).

| | | ST08 | Simulated ST08 | ST09 | ST10 | ST11 | ST12 | ST14 |
|---|---|---|---|---|---|---|---|---|
| BG rate (0.1Hz) | $\overline{ch0}$ | 2800.6 | 2500 | 3915.9 | 2364.5 | 1530.8 | 1501.7 | 1499.4 |
| BG counts (counts/100m) | $\overline{ch0}$ | 0.1935 | | 0.2610 | 0.1582 | 0.1022 | 0.0992 | 0.1000 |
| Remarks | | Nearly full moon | | Icing, full moon | Icing | | | New moon |



## 4.4 Nonlinear effects of photon counting detectors

An ideal detector operated in single photon counting mode should obey a linear behaviour (Donovan et al., 1993). However, nonlinearity can appear in the detector's response, which leads to the ratio of incident and detected photons being dependent on the rate of the incident photons. Above certain rates of incident photons, it becomes increasingly likely that a photon arrives at the detector during the dead time following the detection of a previous photon. As a consequence, the new photon is not detected and the detection efficiency decreases with increasing photon incident rates. The nonlinearity caused by the dead time can become especially noticeable at lower altitudes with larger signals. One way to deal with the deviating number of photon counts due to the dead time is to correct the deviations after the data acquisition (Rapp et al., 2019). The actual number of photon counts $P_R'$ can be derived by:

$$P_R' = \frac{P_R}{1 - P_R * \tau_D} \tag{11}$$

with the detector dead time $\tau_D$. Note that the reason for splitting the signal into different channels is not only given by the desire to increase the measurable dynamic range but also to ensure that the incident photons rates stay approximately 10 times below the maximum detectable rate $\sim \frac{1}{\tau_D}$. In this regime the nonlinearity can be approximated using Eq. (11) and, after correction, linearity cam be assumed.

We already saw that the low channel of ALIMA contains photon counts that reach count rates where a nonlinearity can not be excluded (Fig 3a). In order to determine if the photon counts are indeed significantly influenced by any nonlinear behaviour of the detectors, one needs to discriminate the number of counted photons by the number of incident photons. However, a distinct separation between incident and counted photons of one channel is not possible with the atmospheric measurements by ALIMA (however, the ratio of counted photons from different channels can show a nonlinearity for high count rates as deviation from a proportionality). Nevertheless, we can still draw conclusions about the detector's nonlinearity from the airborne measurements:

The shape of a received pulse of an initially released laser pulse (which in reality cannot be described by a Dirac delta distribution but has a temporal finite width) will be elongated in its temporal dimension due to the different travel times of the backscattered photons from different altitudes. Hence, the counts of the received pulse distribution are more spread towards the tails. Basically, the pulse shape follows the decreasing density with increasing altitude. The counts are further reduced due to the losses by scattering into other directions and the attenuation of the laser pulse. The photon background, however, slightly increases the counts of the received pulse distribution. Within the troposphere and stratosphere, also the atmospheric contribution of the photon background (i.e. moon) might be attenuated by Rayleigh extinction and absorption by ozone. The contribution to the photon background by the dark current should be also constant with altitude since it does not depend on any atmospheric condition. The nonlinearity of the detectors, however, may mimic the altitude-dependent behaviour in the signal because e.g. the effect of heating caused by the current flowing through an APD detector exhibits the same or a larger time constant as the signal responsible for the current. Typically, this effect, if significant, results in enhanced dark count rates at low altitudes which then relax to approximately normal dark count rates at high altitudes. We check whether such a behaviour





is present in the ALIMA data by looking at cumulative histograms (Fig. 11). If the photon background is perfectly constant with altitude, the cumulative histograms should increase linearly. Figure 11 shows that for one temporal bin with 10 s integration period (accumulation of 1000 laser pulses), the cumulative increase of the ALIMA background photons slightly deviates from a linear increase. These deviations are visible in all channels. On average, the ALIMA photon background (from

450 ST08, as well as ST09, ST10, ST11, ST12 and ST14) features a nearly constant increase of photon counts with altitude (not shown) which is however not significant. We conclude that the influence of the detector's nonlinearity in the photon background of the ALIMA measurements is negligible. At lower altitudes with high signal levels, the influence of the dead time on resulting retrieved temperatures is minimized by applying the dead time correction (Eq. 11) to the measured photon counts. Without a suitable correction for dead time effects, the retrieved temperatures can contain biases up to 4 K to 8 K. The

455 uncertainty of the dead time is especially concentrated near the channel transitions at the lower boundary of each channel

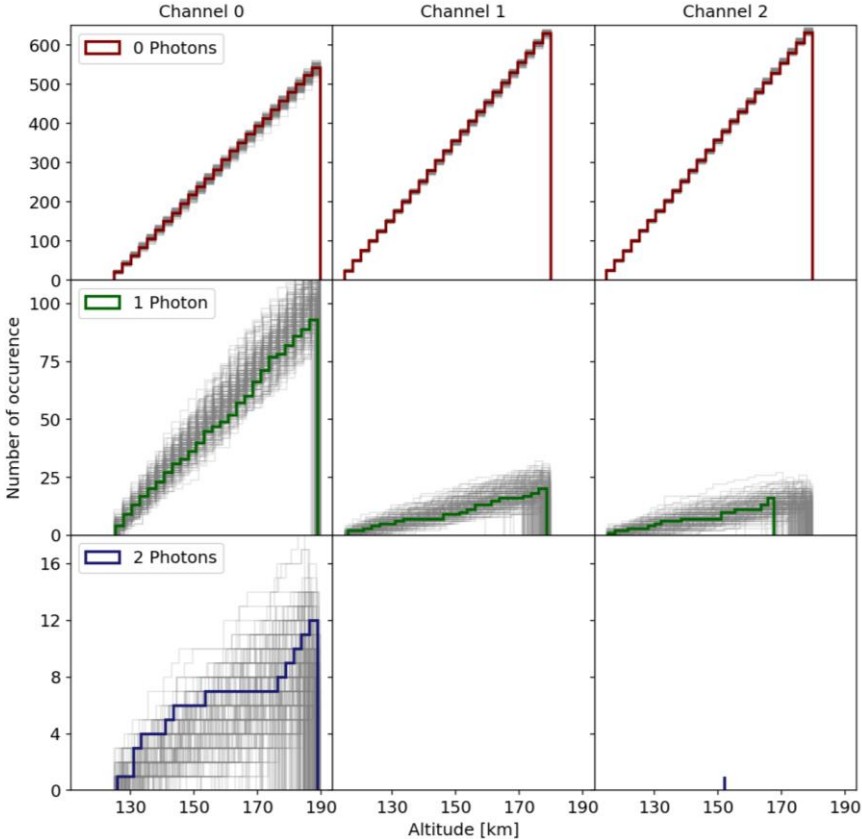

**Figure 11:** Cumulative histograms of ALIMA ST08 background photon counts at 00UTC, binned to 2.5 km altitude intervals for the three channels and categories 0, 1 and 2 detected photons per bin. Grey lines show the results of a Monte Carlo experiment with 200 runs calculating the cumulative histogram of random Poisson distributed numbers with an expected value based on the ST08 flight statistics. If no grey lines plotted: less than 50 Monte Carlo runs indicated photons due to the small sample means.



where the largest photon count rates are measured. The constant photon background also shows that the contribution of the photon noise is insignificant above approximately 100 km. Furthermore, the distribution of ALIMA photon background counts is consistent with that of Poisson distributed random numbers, which verifies that also the counted photons actually follow a Poisson distribution.

## 5 Summary and conclusions

In this study, we analysed different sources of uncertainty in airborne Rayleigh lidar measurements of middle-atmospheric temperature in the range between 20 km to 80 km. The sources are the attenuation of the signal by Rayleigh extinction and absorption by ozone, photon noise, the photon background and the nonlinearity of photon counting detectors. We performed simulations of lidar photon count measurements based on research flight ST08 of the SouthTRAC-GW campaign conducted in September 2019 in Tierra del Fuego, Argentina. The simulations were used to quantify biases and the magnitude of the different contributions to the temperature uncertainty. The results are summarized in Table 4. The uncertainties of the retrieved temperatures are on average dominated by absorption by ozone. Absorption by ozone affects the retrieved temperatures approximately between 25 km to 55 km with an altitude-dependent characteristic defined by the vertical structure of the ozone concentration, resulting in maximum biases of 2.5 K. Uncertainties in retrieved temperature related to attenuation by Rayleigh extinction and photon noise feature magnitudes which are on average similar to the effect of the attenuation by ozone absorption. However, whereas Rayleigh extinction only effects the retrieved temperature below approximately 30 km, the effects of photon noise are largest above 65 km. Moreover, the uncertainty caused by Rayleigh extinction can be significantly reduced by incorporating a suitable correction in the retrieval. The airborne operation of the lidar results in the overall reduction of photon counts by Rayleigh extinction of approximately 2.5 % to 3.5 % compared to a 19 % to 20 % reduction experienced by ground-based lidars. The photon noise can lead to temperature differences of up to ±25 K in extreme cases at high altitudes with small SNR when using small integration periods, as e.g. 1 min and > 65 km. Since the photon noise is always present, this error cannot be prevented in the temperature retrieval of high temporal resolution data, but the resulting uncertainties decrease downwards and become insignificant when averaging over larger integration periods or analysing altitudes of larger SNR. The analysis of ALIMA measurements indicates that uncertainties related to the photon background are negligible in case of night-time operation of ALIMA. The remaining nonlinear effects of the detectors on the retrieved temperatures can be effectively corrected by implementing a suitable dead time correction. Even though the magnitude of the uncertainty related to nonlinearity can be larger than the uncertainty of absorption by ozone, the nonlinearity is significant only in certain smaller altitude ranges where signal rates are very high. This study showed that temperatures can be retrieved from ALIMA photon count profiles with a remaining uncertainty of ≤ 1 K, if all known biases, of e.g. Rayleigh extinction, absorption by ozone and nonlinearity, are suitably corrected.

The study of ALIMA photon counts demonstrates the quality of data acquired by today's generation of lidars. The performance of the ALIMA system leads to determinable sources of uncertainties, which can, therefore, be considered in the temperature





retrieval. The magnitudes of the temperature uncertainty due to the Rayleigh extinction and absorption by ozone demands the inclusion of appropriate corrections in the retrieval. Furthermore, we confirm with our data analysis that the photon background

measured by ALIMA can de facto be assumed to be constant with altitude. A constant background not only simplifies the determination of the background in the retrieval, but at the same time also improves the accuracy of retrieved temperatures.

The airborne operation of a Rayleigh lidar is advantageous as the lidar is closer to the probing volume compared to a ground-based operation and suffers less from Rayleigh extinction. But it also complicates the temperature retrieval due to flight manoeuvres changing the pointing of the laser beam, i.e. during climbs, dives and curves. Therefore, long and straight flight

legs at a constant flight altitude are recommend.

The presented biases in retrieved temperatures influence the temperature mainly in the vertical and to a much lesser extend in time. Temperature perturbations obtained through a temporal filtering at a constant altitude are, therefore, not affected and the resulting uncertainties are not important for gravity wave analyses.

As next steps in the analysis of ALIMA measurements we will evaluate what the smallest temporal and vertical scales are that

ALIMA can reasonably measure and resolve. Emanating from this analysis, we will investigate which physical processes, e.g. turbulence or small-scale secondary gravity waves, can be resolved in the middle atmosphere with ALIMA.

**Table 4:** Contributions of different sources of uncertainty to airborne Rayleigh lidar measurements and their quantitative effects on retrieved temperatures.

| Source | $\Delta P_R$ | Absolute $\Delta T$ | | Correctable | Remark |
|---|---|---|---|---|---|
| | | mean | max | | |
| Seeding error | --- | --- | --- | no | depends on seeding error, any reasonable $\Delta T$ becomes negligible ~ 15 km below seeding altitude |
| Rayleigh extinction | 2.5 % - 3.5 % | 2 K | 2.5 K | yes | Cold bias below 25 km; altitude-dependent |
| Absorption by ozone | 2.5 % - 3 % | 2 K | 2.5 K | yes | Cold bias between ~ 25 km to 55 km; altitude-dependent |
| Photon noise | < 20 % - 30 % (at seeding altitude) | 1 K – 2.5 K | 25 K (at seeding altitude) | no | Cold bias at high altitudes for small integration periods (1 min) |
| Photon background | < 1 % - 2 % (at seeding altitude) | --- | < 1 K | (yes) | Influence decreases exponentially downwards |
| Nonlinearity (detector dead time) | --- | 2 K | 4 K – 8 K (above channel transition) | yes | Occurrence and peculiarity depend on the rate of incident photons; greatest at lower boundary of each channel |




*Code and Data availability.* ECMWF ERA5 data can be freely accessed from
https://www.ecmwf.int/en/forecasts/datasets/reanalysis-datasets/era5. ALIMA data can be downloaded from the HALO data
base https://halo-db.pa.op.dlr.de/.


*Author contribution.* SK performed the lidar simulations and data analysis and prepared the manuscript. BK built and operated
ALIMA, wrote the retrieval algorithm and assisted with the paper preparation. MR supervises the PhD work of SK and assisted
with the paper preparation.

*Competing interests.* The authors declare that they have no conflict of interest.

*Acknowledgements.* This work was partly funded by the German Federal Ministry for Education and Research under grant 01
LG 1907A (project WASCLIM) in the frame of the Role of the Middle Atmosphere in Climate (ROMIC)-program. Further
support for the SOUTHTRAC mission came from internal funds of the German Aerospace Center, the Karlsruhe Institute of
Technology, Forschungszentrum Jülich, and the German Science Foundation (DFG) through the HALO-SPP (DFG-project
number 316646266). Further support by DFG under grant PACOG/RA 1400/6-1 in the frame of the DFG-research group MS-
GWAVES is also acknowledged. Access to ECMWF data was granted through the special project "Deep Vertical Propagation
of Internal Gravity Waves".

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
