# Peer review of "Estimating the uncertainty of middle-atmospheric temperatures retrieved from airborne Rayleigh lidar measurements"

_Atmospheric Measurement Techniques, 2021_

## Author Comment (AC1)

**Letter to the Editor**

For manuscript "Estimating the uncertainty of middle-atmospheric temperatures retrieved from airborne Rayleigh lidar measurements"
(https://doi.org/10.5194/amt-2021-310)

Dear Prof. Sica,

Many thanks for your efforts as editor of our manuscript submitted to Atmos. Meas. Tech.
We have carefully read the reviews and appreciate all the comments and suggestions. As you already pointed out, we missed citing and discussing major literature for the topic of our manuscript. We will incorporate the necessary references. Moreover, we realized that we have not adequately presented the purpose of our manuscript. In the following, we respond to the individual comments of the two anonymous referees.

With kind regards,
Stefanie Knobloch
(on behalf of all co-authors)

**Comments on Review of Anonymous Referee #1:**

Dear Referee #1,
Thank you very much for your detailed review on our submitted manuscript. Your review helped us very much to see the weaknesses in the manuscript and to understand what causes them.

**(i):**

"This paper focuses on the retrieval of middle atmospheric temperatures from the backscatter signals of an airborne Rayleigh lidar system. The technique was introduced forty years ago by Hauchecorne and Chanin (1980). It has been in operational use for more than 30 years at a number of stations, e.g. within the Network for the Detection of Atmospheric Composition Change (http://www.ndacc.org). Essentially, the paper re-iterates data analysis considerations that have become standard for many years. So nearly all results presented in the paper have been well known in the scientific community: Ozone absorption needs to be accounted for, or will introduce a bias (Leblanc et al. 1998; Sica et al. 2001). Uncertainties introduced by photon noise, background light, counter non-linearity, Rayleigh extinction are all well known and extensively discussed, e.g., by Leblanc et al. (2016). Lidar return signals have been simulated, and retrieval algorithms have been tested by Leblanc et al. (1998, 2004). Modern optimal estimation algorithms for temperature retrieval from Rayleigh lidar signals are given by Sica and Haefele (2015), and Jalali et al. (2018)."

Indeed, the analyzed impact of all above mentioned uncertainty sources are already presented in various papers. Firstly, we missed the references and discussion of some of them in our manuscript. We realize that a broader discussion may result in substantial improvement of the manuscript. Secondly, our attempt is not to compete against those papers but to compare the analysis of measurement uncertainties tailored to our airborne lidar to well-known measurement uncertainties of ground-based lidars. The review shows us that we have not clearly explained our goal and did not implemented our analysis well. The primary goal of our study is to demonstrate and analyze the effect of the airborne operation of a lidar on the measurement performance Though many aspects and sources of uncertainties were discussed in previous publications, none of these publications include the application to airborne lidars. In comparison to ground-based systems, the operation of a lidar on a fast-moving platform results in additional complications and peculiarities in the performance analysis of such a lidar system. The discussion of these issues is our goal.

**(ii):**

"I did not understand the attempt to quantify signal-induce-noise (SIN) in section 4.4. I am not sure if SIN can be detected in this way. A more standard way would be to look at the decay of the background for varied maximum light exposures of the photo-detector."

Our aspiration is to analyze the possibility of a nonlinear behavior of a detector based solely on existing measurements that were obtained during the SouthTRAC campaign. Your proposed method implies the need for additional measurements carried out in the laboratory.

**(iii):**
"Unfortunately, overall, I see very little new scientific findings in this paper, and feel that very major revisions are needed, before this is acceptable for publication in AMT."

Yes, we agree with your conclusion. In the present version, the conclusions of the manuscript are not sufficiently worked out and guide to a different statement than we actually want to make. We will reorganize the manuscript and clearly state the purpose, namely, the investigation of differences between our presented uncertainties of an airborne lidar and uncertainties of a ground-based lidar (e.g. based on literature).

**(iv):**
"The comparison of observed, simulated, and ERA-5 analysed temperature waves in Fig. 4 is new and interesting. It shows the capabilites of the airborne lidar to observe large gravity waves over one of the worlds hot-spots. This is worth presenting, maybe expanding."

We agree that the comparison represents an engaging result which we should give more attention, especially for the sake of future gravity waves analyses based on ALIMA measurements.

**(v):**
"The claim that the skewed Poisson statistics of photon counting introduce a high bias in retrieved density and a low bias in retrieved temperature is new to me, but not explained very clearly. I would expect that this bias would be within the estimated uncertainty, and would only be significant at very low total photon counts. However, it would be interesting to have a more detailed look at this."

We are not sure what you are actually referring to, e.g. which paragraph/chapter, Fig. 7?

**(vi):**
"In summary, I suggest major revisions for this paper. All the parts that re-iterate well known facts, already described in, e.g., Leblanc et al. (2016) should be dropped, or should be shortened to a few paragraphs. The focus of the paper should be on the new findings, e.g. those I mentioned above. Most AMT readers have little time, so new papers need to be concise and need to present important new findings only."

We agree with you! We missed to adequately state the purpose of our manuscript And focused too much on the description of our work instead of discussing the new findings as well as their implications for the analysis of airborne lidar temperature profiles.

**Comments on Review of Anonymous Referee #2:**
Dear Referee #2,
Many thanks for your mindful review and pointing out the weaknesses of our manuscript.

**(i):**
"This manuscript aims at characterizing measurement uncertainty for an airborne Rayleigh lidar (ALIMA), looking upward towards the stratosphere and mesosphere. One of the science objectives is to observe density and/or temperature disturbances associated with the propagation and dissipation of gravity wave in the middle atmosphere. The authors use lidar signal simulation to estimate certain components of this uncertainty. Most of the manuscript repeats what has been already published, and so my main recommendation is to re-submit after major revisions, including a re-organization of the manuscript to re-balance the weight given to each section, based on what has been already published and what has not. I recommend to refer to Leblanc et al. (2016) (citation below) who provide, for example, quantitative estimates of the uncertainty associated with molecular extinction and ozone

absorption (this part should be straightforward and not exceed a paragraph or two in the revised manuscript)."

Yes, we agree with your recommendation.We will restructure the manuscript and highlight the peculiarities which arise due to the airborne operation of our lidar.

**(ii):**

"Unfortunately, the manuscript suffers from a major mistake in the quantification of the temperature correction associated with ozone absorption. If I am not mistaken, their ozone optical depth and ozone absorption correction were computed using O3 mixing ratio rather than O3 number density, which explains why they found a maximum impact at 35 km rather than 22-24 km. Fig 7 (left) of Leblanc et al. (2016) and Figs. 4 and 5 of Sica et al. (2001) both show a maximum impact in the lower stratosphere associated with O3 ND peaking at 23-26 km."

You are absolutely right. In our code we do calculate the O3 number density but unfortunately assign the wrong variable in the calculation of the optical depth afterwards. We already corrected the mistake and will update our analysis. Thank you for pointing that out!

**(iii):**

"I also strongly recommend that the authors make a clear distinction between what is uncertainty, error, and bias, which eventually, will greatly help them re-shape the manuscript towards a well-defined objective. I believe the current objective of the authors is to assess the quality of the ALIMA measurements, and eventually provide a full uncertainty budget. Lidar simulation is not needed for most of this estimation work. Some of the figures shown in past publications can serve as guidance to present their results in the revised manuscript. Here are suggested definitions that might help re-focusing the next manuscript: Bias = a value, negative or positive, describing an observed, systematic (i.e., repeatable) difference between 2 observations. Error = A value, negative or positive, describing the actual (unknown) difference between the true value and the measured value. Uncertainty = A value, always positive, describing statistically the best estimate (or magnitude) of the (unknown) error arising from a specific physical effect or retrieval approach that drives the final, reported value away from its true value. For example, "temperature uncertainty due to ozone absorption" is an estimate of the error due to the fact that the ozone absorption is not perfectly accounted for in the temperature measurement/retrieval. Unlike error and bias, uncertainty is a controlled quantity."

Thank you for your definitions of error, bias and uncertainty. We agree that we have not clearly defined these terms and may have used them inconsistently within our manuscript. Following your definition, we want to characterize the error in temperature profiles retrieved from ALIMA measurements. This is also the reason why lidar simulations are needed. We have a known atmospheric state (ERA5 data) and can analyze how the retrieved temperature deviates from the known state and what error sources contribute to these deviations. We will revise the manuscript according to your suggestions.